# Beyond high hopes: A scoping review of the 2019–2021 scientific discourse on machine learning in medical imaging

**Vasileios Nittas**[1,2], **Paola Daniore**[3,4], **Constantin Landers**[1], **Felix Gille**[3,4], **Julia Amann**[1], **Shannon Hubbs**[1], **Milo Alan Puhan**[2], **Effy Vayena**[1], **Alessandro Blasimme**[1] *

1 Health Ethics and Policy Lab, Department of Health Sciences and Technology, Swiss Federal Institute of Technology (ETH Zurich), Zurich, Switzerland, 2 Epidemiology, Biostatistics and Prevention Institute, Faculty of Medicine, Faculty of Science, University of Zurich, Zurich, Switzerland, 3 Institute for Implementation Science in Health Care, Faculty of Medicine, University of Zurich, Switzerland, 4 Digital Society Initiative, University of Zurich, Switzerland

* alessandro.blasimme@hest.ethz.ch

**Data Availability Statement:** All relevant data are within the manuscript and its Supporting Information files.

## Abstract

Machine learning has become a key driver of the digital health revolution. That comes with a fair share of high hopes and hype. We conducted a scoping review on machine learning in medical imaging, providing a comprehensive outlook of the field's potential, limitations, and future directions. Most reported strengths and promises included: improved (a) analytic power, (b) efficiency (c) decision making, and (d) equity. Most reported challenges included: (a) structural barriers and imaging heterogeneity, (b) scarcity of well-annotated, representative and interconnected imaging datasets (c) validity and performance limitations, including bias and equity issues, and (d) the still missing clinical integration. The boundaries between strengths and challenges, with cross-cutting ethical and regulatory implications, remain blurred. The literature emphasizes explainability and trustworthiness, with a largely missing discussion about the specific technical and regulatory challenges surrounding these concepts. Future trends are expected to shift towards multi-source models, combining imaging with an array of other data, in a more open access, and explainable manner.

## Author summary

Machine learning is becoming an important part of digital health and medical imaging. Many believe that it is the solution to some of the challenges our medical systems currently face. In this study, we reviewed the literature to explore this topic, focusing on the promises, challenges, and future developments. The literature emphasises that machine learning allows us to use medical images in ways that are more reliable and precise, requiring less time and resources. That can lead to better decision-making, as well as allow for more people to access affordable image-based care. Some of the mentioned challenges include the large differences between images and imaging techniques, the difficulty in accessing enough high-quality images, the costs and infrastructure associated with that and the resulting geographic inequalities. In addition, the literature emphasizes that is difficult to understand how machine learning works, as well to assess how valid and reliable

**Funding:** AB and EV financed this project through the Swiss National Science Foundation National Research Program 77 (NRP77) Grant (407740_187356). The funders had no role in study design, data collection and analysis, decision to publish, or preparation of the manuscript.

**Competing interests:** The authors have declared that no competing interests exist.

it is in analysing medical images. Equally difficult is its regulation and integration in everyday clinical work. The future is expected to bring machine learning models that will be able to analyse different types of images and other clinical data at once, in ways that are more transparent and understandable.

## Introduction

Amidst the so-called digital health revolution, the Lancet and Financial Times Commission on governing health futures 2030 called upon decision-makers and experts to consider digital technologies as modern health determinants [1]. Although the digitalization of healthcare lags behind other industries, the amount of healthcare data generated electronically is now larger than ever [2]. Harnessing the power of that data leaves no alternative besides the use of technology [2]. Technologies such as artificial intelligence (AI) and machine learning (ML) have rapidly become the epicenter of that revolution, as well as of political, scholarly, and public attention [2,3].

Much of the hype around AI and ML is rooted in mainstream media, with claims about the limitless capabilities of data-driven algorithms to achieve disproportionate reach [4–7]. Headlines with terms such as "transforming healthcare" and "reshaping medicine" generate unrealistic expectations, and inevitably skepticism by healthcare professionals and patients alike [7,8]. AI is widely used as a broad umbrella term, rooted in computer science and describing computational methods that allow computers to perform tasks that would traditionally require human intelligence [9,10]. While often used interchangeably with AI, ML is a subset of AI which allows computers to self-sufficiently learn to perform a pre-defined task and possibly improve over time [11].

Medical ML undeniably achieved remarkable progress, particularly in the field of visual pattern recognition [2,12]. ML and its subsets such as deep learning (DL) have been extensively applied in the field of medical imaging, accounting for about 40% of all ML-related publications in the health sciences domain over the past five years. With the amount of imaging data skyrocketing, ML offers efficient ways to use this data for clinical decision-making, ranging from computer-aided diagnosis to radiomics and image-guided therapy [13]. Considering these successes, together with the yet-undefined boundaries of the discipline, uncertainty about the clinical impact of AI is expected [5,7].

It is thus appropriate to ask: what are the prospects of ML to become clinically integrated, and what impediments can slow down or stifle innovation? To go from proof-of-concept models built on retrospective data, to ML systems capable of improving healthcare, the nature of the expected impact of AI in healthcare must be clarified. Moreover, we need to identify perceived bottlenecks and impediments to the successful integration of ML in routine clinical medical imaging practice. Such information is key to inform policy development around a clear vision of AI-driven transformation in the medical imaging sector and to foster the emergence of efficient technological and clinical validation standards.

To this aim, this review explores and critically discusses the latest (2019–2021) scientific discourse around ML in clinical imaging, highlighting the field's promise, challenges, ethical and regulatory implications, and future directions. We focused on the big five chronic conditions, including cardiovascular diseases, diabetes, stroke, chronic respiratory diseases, and cancer. The rationale behind this review's focus is the following. First, we believe that the big five chronic conditions constitute a significant amount of available scientific output, and thus, provide a representative picture of current trends. Second, focusing on a selected number of medical areas ensures that this review's output remains manageable. Third, these five chronic

conditions are responsible for a considerable share of healthy years lost, globally. These areas are therefore some of the most relevant for ML application and impact. To the best of our knowledge, this is the first systematic approach to exploring the scientific discussion around ML in image-based care, with a comprehensive outlook encompassing scientific and socio-technical aspects.

## Methods

We conducted a scoping review of narrative reviews and editorials. These two types of publications typically reflect expert opinion about the latest trends and prospects of any given scientific field. Therefore, they are likely to provide a broad picture of current scientific discourse, being published in a shorter timeframe not constrained by complex and lengthy methodologies. Our approach was guided by Arksey and O'Malley's framework, as well as Levac, Colquhoun, and O'Brien's conceptual extensions [14,15].

### Search strategy and selection criteria

Guided by the study's aims, a specialized librarian developed a comprehensive search strategy that was applied on six electronic databases, including Medline, Embase, CINAHL, PsycInfo, Scopus, and Web of Science. Searches were run on August 12, 2020, and updated on December 5, 2021, to cover publications until August 31, 2021. We used multiple variations (and synonyms) of the following search terms: machine learning, artificial intelligence, and medical imaging. We only included narrative reviews, editorials and commentaries, written in English. An example of our search strategy (Web of Science) is provided in S1 File. In addition, within the same time frame, we hand-searched four of the most prominent journals in the field, including the Journal of Medical Internet Research, npj Digital Medicine, Lancet Digital Health, and Nature Machine Intelligence.

We conducted title, abstract, and full-text screening in duplicate and independently, guided by a set of predefined exclusion criteria (see Table 1) and excluding papers if any of the below criteria were true. Any disagreements were resolved by a third reviewer. For title and abstract screening, we utilized two web-based systematic reviewing tools, DistillerSR (Evidence Partners) and Rayyan [16,17].

### Data extraction

We extracted (single reviewer) data with a predefined data extraction sheet that captured the study's aims, including (a) strengths and promises of ML use in clinical imaging, (b) the

**Table 1. Selection Criteria.**

[1] Addressing ML (and sub-fields, e.g., deep learning) AND Imaging. Studies using the term "AI" interchangeably with ML are included.
 ◦ ML & Imaging should be addressed either in (a) title and/or (b) abstract AND throughout the main body
[2] A narrative review, commentary or editorial
[3] Focused on one (or multiple) of the five chronic conditions:
 - Cardiovascular diseases
 - Cancer
 - Chronic Respiratory diseases
 - Diabetes
 - Stroke
[4] Not a technical paper (e.g., describing algorithms without conveying a scientific opinion, neither discussing the broader clinical, social, ethical, or regulatory implications of ML)
[5] Published after the first of August 2019 and before the first of September 2021
[6] Written in English

challenges (including weaknesses, barriers), (c) the potential solutions to these challenges, and (d) the field's future trends. We initially tested the form on a sample of 15 studies, which allowed for further refinement and adjustment. Finally, we validated the data extraction sheet through multiple rounds of internal reviews.

### Data synthesis and reporting

We synthesized our findings following a qualitative and iterative thematic approach, conducted by one reviewer and quality-checked by a second reviewer [15]. After data familiarization, we generated initial codes inductively. We then clustered emergent themes and synthesized these in conceptual maps that provided an overall picture of our findings. The reporting of our results was based on the Preferred Reporting Items Extension for Scoping Reviews (PRISMA-ScR) statement [18].

## Results

Our database searches yielded a total of 8079 references, and our hand searches 23. Of these, 7249 were excluded at the title and abstract screening. The full texts of 673 papers were assessed, leading to 561 further exclusions. We finally included 112 publications, listed in S2 File. Fig 1 provides the PRISMA flow chart for our screening and study inclusion process [19,20].

Most included publications (published between August 2019 to August 2021) were narrative reviews (n = 92, 82%), followed by editorials and short opinion-based papers (n = 20, 18%). First authorship was affiliated with institutions in the European region in 48 (43%) publications, followed by 38 (34%) from North America, 23 (20%) from Asia, and three (3%) from Australia. Most publications addressed the use of image-based ML in oncology (n = 51, 45%), followed by cardiology (n = 39, 35%), respiratory diseases (n = 5, 4%), diabetes (n = 3, 3%) and stroke (n = 3, 3%). Eleven (10%) studies had a multi-disease focus. The following paragraphs provide our findings around: (1) the strengths of ML use in medical imaging, (2) the challenges of ML use in medical imaging, with a focus on ethical and regulatory challenges, (3) the potential solutions to these challenges and (4) future trends.

About 60% (n = 66) of all included publications primarily focused on a single and 40% on multiple imaging modalities (n = 46). Among these, computed tomography (CT) was the most commonly addressed (n = 46, 41%), followed by magnetic resonance imaging (MRI) (n = 33, 29%), ultrasound imaging (n = 20, 18%), nuclear imaging (n = 22, 20%), digital pathology (n = 11, 10%), X-ray imaging (n = 8, 7%), and retinal fundus imaging (n = 3, 3%). About 20% (n = 22) of included publications primarily focused on the use of ML for classification tasks, most often employing artificial neural networks, convolutional neural networks and support vector machines. Among those, most publications focused on cancer (n = 10), followed by cardiovascular diseases (n = 6), pulmonary diseases (n = 4) and diabetes (n = 2). About 14% (n = 16) of publications focused solely on prediction and regression, all of them addressing either cancer (n = 10) or cardiovascular diseases (n = 6). The remaining 66% (n = 74) of publications simultaneously addressed both classification and prediction.

### Strengths/Promises

Almost all publications (n = 110, 98%) reported at least one expected strength of ML applications in imaging (with two not mentioning any). We identified four closely related but distinct domains: (a) analytic power, (b) efficiency, (c) clinical impact, and (d) equity. Each of these contained subdomains, described in Table 2.

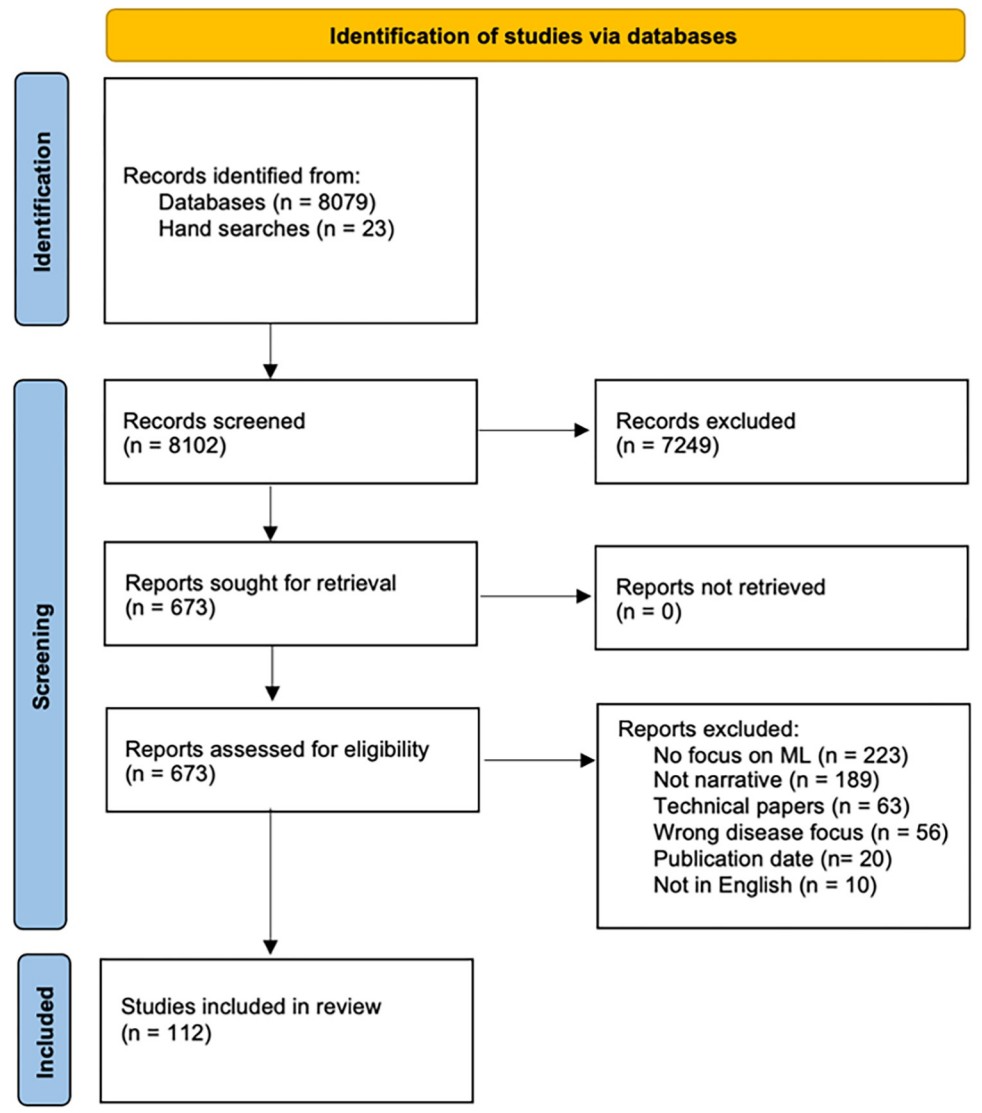

**Fig 1. PRISMA flowchart**

## Challenges/Impediments

Most publications (n = 102, 92%) reported at least one challenge of ML applications in medical imaging (with ten not mentioning any). We distinguish between those challenges mostly related to imaging, and those generally pertinent to ML. The first are summarized in Table 3, the second in Table 4. The ML challenges were further divided into three distinct domains, including (a) structural barriers, (b) validity and performance, and (c) clinical relevance.

The challenge of image heterogeneity was mostly discussed in the context of cancer and cardiovascular diseases [9,21–29]. This is particularly applicable to imaging modalities with no standardized pixel values or agreed format, such MRI and digital pathology imaging, or those with high noise and low resolution, such as intravascular ultrasounds [23,30–32]. In both cases, interobserver reliability is high and feature extraction challenging [31,33]. Besides the inevitable heterogeneity arising from the use of different image generation, reconstruction and preprocessing techniques, there is a large impact by confounders, such as noise due to artifacts

**Table 2. Reported (expected) strengths and promises of image-based machine learning.**

| Domain | Subdomain(s) | Descriptions |
|---|---|---|
| Analytic power | accuracy | *ability to classify images as well as experts, ensuring diagnostic and prognostic accuracy* |
| | superiority | *ability to differentiate images better than an expert, identifying patterns not always visible to the human eye* |
| | objectivity | *lack of human subjective biases and errors, reducing variability and improving comparability and reproducibility* |
| | big data use | *ability to handle and analyse large amounts of data and tackle the challenges of big data* |
| Efficiency | time efficiency | *ability to speed up analysis and clinical translation through automation of otherwise time-consuming manual tasks* |
| | cost efficiency | *ability to lower direct and indirect costs through time and diagnostic efficiency, automation, and enhanced workflows* |
| Clinical impact | workflow improvements | *ability to optimize clinical workflows through integrating, automating, streamlining, and structuring processes* |
| | decision making | *supporting faster, cheaper, more accurate and higher-level decision making and clinical interpretations* |
| | personalization | *facilitate personalized care, though higher analytic power, efficiency and improved clinical workflows* |
| Equity | reach, access, and affordability | *promise of increased geographic reach of and better access to affordable image-based healthcare* |

(e.g. implants in CT images) or motion, which is especially relevant to heart imaging [22,29,34]. Authors mentioned that heterogeneity increases as the dimensionality, number and complexity of images increases, making it hard for features to be extracted and used for ML training purposes [28,35]. Modalities such as echocardiography create large datasets, with varying operator-depending image quality, yielding high complexity and interrater variability [36,37]. This is applicable to cardiac imaging in general, which is dynamic and generated while the heart is in constant motion, posing particular challenges for ML [36,37] Variation, noise, and complexity ultimately hinder the extraction of information from images, as well as the determination of ground truth, which is essential during training and validation of ML algorithms [9,27,38].

The next four challenges of image and annotation scarcity, class imbalance, image silos and cost all fall under the category of missing resources and infrastructures in medical imaging. Image scarcity was primarily mentioned in the context of neuro-oncology, particularly with

**Table 3. Image-related challenges of machine learning, as reported in the literature.**

| Domain(s) | Descriptions |
|---|---|
| Image heterogeneity | *high image variation due to non-standardized acquisition, reconstruction and preprocessing protocols* |
| Image scarcity | *image scarcity, especially in complex fields such as neuro-oncology and rare diseases* |
| Annotation scarcity | *lack of properly annotated image datasets, and if annotated, subject to high inter-rated variability* |
| Class imbalance | *datasets often include too few cases (images indicating disease)* |
| Image silos | *large volumes of imaging data exist in silos, with low interconnectivity* |
| Cost | *access to medical images often costly, requiring adequate training and infrastructural support* |
| Subjectivity | *image selection, segmentation and annotation conducted by human experts and not entirely free of subjective biases* |
| Inequities / bias | *most advanced imaging modalities mostly accessible in high-income countries and to high-income population subgroups, leaving many underrepresented* |

**Table 4. General machine learning challenges, as reported in the literature.**

| Domain | Subdomain(s) | Descriptions |
|---|---|---|
| Structural barriers | assessment | *challenging validation, evaluation, and prediction of algorithm performance* |
| | regulation | *lacking regulatory and reimbursement frameworks, inhibiting standardization and creating liability uncertainties* |
| | resources | *algorithm development requires extensive time, financial, and human resources* |
| Validity and performance | generalizability | *training with small samples and lacking external validation, limiting broader applicability* |
| | reliability | *training with small samples and lack of testing, leading to often overfitted algorithms, uncertainty, and poor performance* |
| | bias | *prone to systematic biases (e.g., learning bias, social biases)* |
| | narrowness (task) | *limited to solving isolated tasks, unlike human intelligence and expertise, which can grasp clinical complexities in full* |
| | narrowness (data) | *most algorithms not capable of integrating multiple data sources simultaneously (e.g., images and health records)* |
| Clinical relevance | explainability | *inherently complex, with data-driven, non-intuitive features, limited understanding of the logic behind algorithms' decisions* |
| | integration | *lacking clinical integration due to structural barriers, uncertainty, missing training, low explainability and trust* |
| | human factors | *human factors (e.g., cognition, biases, demographics, experience) mostly overlooked in the design and development of ML algorithms* |
| | clinical performance | *most current studies retrospective and "offline", not providing evidence on the clinical performance of algorithms in multidimensional and dynamic settings (e.g., real-world conditions)* |

very heterogeneous tumours like Glioblastoma Multiforme, as well as in the context of nuclear imaging [21,39]. Authors suggested that where images are abundant (e.g., in areas such as breast imaging), they often lack adequate annotations, as these are often not required for regular clinical workflows [40,41]. On the opposite, detailed enough annotations depend on scarce resources such as time, effort and training [40,41]. Some authors mentioned that a plethora of images are stored in silos, across different institutions and platforms, making access and interconnectivity difficult [42,43]. Within these silos, the number of images from healthy individuals is often significantly larger than the number of images indicating disease [41,44]. In addition, when disease is indicated (e.g. malignant nodule) the occupied region of interest is often relatively small and hard to detect [24]. This class imbalance makes a dataset less valuable for ML training and was mentioned in the context of cancer and diabetic retinopathy [41,44]. Primarily mentioned in the context of stroke, images alone are often not powerful enough for creating clinically useful ML tools. Predicting or classifying conditions like stroke depends on an array of contextual data, which images alone do not provide [45]. Finally, accessing medical images to train ML algorithms can be a costly endeavour, requiring large capital costs, skilled and trained personnel and adequate infrastructures [46]. This is particularly relevant to the generation and use of digital pathology images, as the digitalization of pathology workflows and the transition to whole-slide imaging comes with particularly high costs [47].

The two final mentioned challenges are those of subjectivity and inequities. Considering that images are often selected, segmented, interpreted and annotated by individuals, subjectivity and bias cannot be entirely ruled out [48]. In addition to that, advanced imaging modalities are mostly accessible in high-income countries and among higher income population subgroups. This generates image datasets that underrepresent low- and middle-income countries and population subgroups from lower socio-economic strata [49]. A strong example of such inequities is in the field of cardiac imaging, with newest imaging technologies often being geographically and socio-demographically limited [49].

Many of the challenges listed in Table 4 have an ethical and/or regulatory dimension. Authors argue that lack of regulatory frameworks limits fairness and increases the risk of biased algorithms and widening inequities [4,37,50–54]. Such inequities are further exacerbated by ML's costs (e.g. technology, training, staff), which limit its use to high-resource settings [55]. Liability was mentioned as a further challenge. The main question posed was: in

case of harm (e.g., due to misdiagnosis), who is responsible? The physician, institution, or algorithm developer? [56–59]. Some argued that ML applications are merely assistive, and thus liability falls with the treating physician [49,58,60]. Others emphasized that many factors (e.g., integration of a tool within an institution or how it changes with use) make liability difficult to assign, further limiting trust and hindering clinical uptake [40,49,61].

The data-dependency of ML algorithms comes with concerns around privacy, security, and data ownership [25,46,50,59,62,63], including secure storage and sharing of data, and protection from data breaches and attacks [54,64]. Authors argue that ML's need for large volumes of images for training and validation poses security and consent challenges. Individualized consent is rarely feasible, creating the need for alternative solutions, such as opt-out consent or broad consents, in which individuals consent to multiple secondary uses of their data, balancing between data availability and privacy [49,54,65]. Even after consent is obtained, the anonymization of images (to which identifiers are often attached) poses a significant processing burden and risk for re-identification [4]. Finally, the question arises of whether patients should be given the option to not include ML in their care [58,66].

Authors also argue that low explainability could reduce trust, in turn hindering clinical acceptance and uptake [9,32,49]. That may potentially affect the patient-physician relationship, interfering in decision-making and eventually leading to unease or confusion [42,56,67]. An additional challenge is the perception of competition between ML and human intelligence, with healthcare providers fearing gradual replacement, although this was often debunked as unlikely [27,59,68,69]. As overreliance on ML can lead to automation bias, increase mistakes, cause avoidable harm, and deskill physicians [54], clinicians should view ML tools as systems to support rather than replace their decision making.

## Suggested solutions for identified challenges

Around 80% of publications (n = 81) provided potential solutions to these challenges. Some argued for facilitating access to high-quality data and existing algorithms, pooling of resources, and dialogue between stakeholders, while ensuring that privacy, ethics, and commercial interests are aligned and safeguarded [30,50,70–73]. Others called for more standardization, including well-defined imaging protocols and research methods (e.g. image acquisition), as well as open access and easily available algorithm codes, to enable reproducibility [9,21,51,74,75]. Standardizing how and what information is extracted from images for ML purposes is another proposed solution [22]. Authors mentioned the Image Biomarker Standardization Initiative, which provides guidelines on how that is to be achieved, yet these remain to be adjusted and adapted to the needs of more challenging imaging areas, such as cardiac imaging [9,71]. The exclusion of image features that are too sensitive to variation has been another suggested solution towards increased standardization [9]. For generalizability, authors emphasized the need for more and better-performed external validation, using multicentric imaging datasets obtained under real-world conditions and representative of the target patient group [4,9,11,22,28,46,72,76–78]. Transfer learning, crowd-sourced, and open-access image repositories, and mathematically generated datasets are alternatives if data scarcity cannot be solved [41,71,79–81].

Beyond enhancing data availability and quality, Radakovich and colleagues highlight the risk of overfitting, which can be regulated through penalization of very complex models or image augmenting techniques that increase heterogeneity (e.g. adding image noise) in datasets with which ML algorithms are trained [80]. A further approach to improving reliability involves training algorithms away from strict outcomes (e.g. disease is/is not present) towards outcomes that communicate uncertainty, a third "grey zone" category [82]. Class imbalance

issues should be addressed with more research into advanced deep learning algorithms that can deal with a low number of cases (images indicating disease) [41]. Some authors pointed to regulatory frameworks as essential for ensuring safe and fair ML; yet they lag behind innovation, and are challenged by the continuously evolving nature of ML algorithms [4,49,54,60,83]. Malpractice insurance must be adjusted to the unique liability challenges posed by AI and ML, complemented by comprehensive risk reduction plans [54].

Acknowledging the black box nature of ML, some authors argued that data science and advanced statistical and computational methods should be integrated into the education of medical professionals [58,59,84–87]. Similarly, ML developers should be educated on clinical needs and challenges, including clinical workflows, so that products avoid unnecessarily delaying clinical practice, overburdening practitioners, or negatively affecting the patient-physician relationship [50,54]. Engagement was repeatedly mentioned, including meaningful inclusion of clinicians, patients, and other stakeholders in the development of ML algorithms [22,39,49,57,88,89]. Finally, some authors argue that ML should be accompanied by understandable feedback on why a decision or classification was made [34]. For example, saliency heat maps visually indicate which parts of an image contribute to an ML decision, confidence intervals communicate uncertainty, and interactive dashboards help physicians better understand a system [12,24,32,40,42,61,90].

## Clinical utility and integration

Some publications included in this review highlighted that clinical integration has not yet been achieved, and thus the clinical utility of ML is limited [9,29,91]. The authors suggested potential reasons for this. First, most ML studies fail to compare their algorithms to current gold standards and instead retain a technical, proof-of-concept focus [9,30]. Others argued that the performance of most ML algorithms is not assessed based on their impact on disease outcomes, quality of care, or cost-effectiveness, with most studies adopting rather non-transparent workflows that do not resemble real-word settings [24,36,38,42,50,76,87,92]. Thus, it remains unknown how many of these tools perform under complex and unpredictable conditions, or how they affect the broader planning and provision of care [40,42,59].

To address this, some authors suggest large, multi-center prospective trials and quasi-experimental studies, embedded in clinical settings and focusing on downstream impact, including patient outcomes, safety, efficacy, and acceptance [36,50,63,78,89,93]. These shall be complemented by economic evaluations, evaluating cost-effectiveness relative to standard care [40,50]. Furthermore, it is essential to account for the human factor when developing and evaluating ML tools, understanding the impact of algorithms on clinical behavior, interactions between physician and machine, and effects of human traits (e.g., cognition, bias, demographics, experience) on algorithm performance [53,64,94]. Finally, authors raised concerns that ML studies are not driven by clinical relevance but rather by the availability of datasets, with little effort to address the black box nature and bias-related challenges of ML algorithms [35,42,84].

## Future directions

One often discussed future trend was the capability of ML algorithms to combine multiple data sources [21,71,81,85,86,95–97], including biological and biochemical data, genomic information, socio-demographics, life-style risks, linkages to electronic health records, and other clinical information [9,22,23,34,68]. Multi-data ML (or multi-omic ML) is expected to inform and improve clinical workflows [21,22,36,42,68,98,99]. Mobile and wearables devices and sensors generate large volumes of yet-untapped health information, and are expected to

democratize and contribute significantly to multi-omic ML, reducing costs and accessibility [35,57,63,69,96,100]. Yet, the actual application of multi-omic ML remains limited, with first efforts being reported in the field of oncology and cardiac imaging (e.g., combining echocardiographic and other clinical data for heart failure diagnosis; combining radiological tumour data with an array of physiological and genomic data) [69,95].

While advocating for multi-data ML, the literature does not provide insights into which types of image modalities yield highest compatibility with physiological and other clinical data. In fact, most publications did not focus on a single imaging modality but imaging in general. This was followed by publications on CT, MRI and echocardiography. On the other hand, the need to combine images with other data might arise from the area of application, instead of the imaging modality per se. Using ML for stroke detection is a good example of that. Decision support in stroke strongly depends on subtle clinical findings and multiple contextual data. For an ML tool to provide value in stroke care, it must be trained with such rich contextual and clinical data, which includes images and physiological data [45]. Similally, to fully and better understand the pathological mechanisms of certain tumors might require ML models that take into account an array of data, combined with images [22].

Another desired trend is that of higher standardization of image generation and processing, as well as multistakeholder engagement [11]. The future should see an emphasis on collaboration between academia, industry, and healthcare institutions to facilitate knowledge exchange and data sharing around AI and ML [11]. For ML tools to be clinically useful, their future use must be rooted in clinical need, while clinicians of the future will, in turn, need to understand the basics of ML and data science [36,45,57,101]. Shared, open-access databases for ML training and comparison are essential to the field's future [31,60].

Finally, the development of robust and less labour-intensive ML, shifting from hand-engineered input to algorithms that automatically extract required information, was another major trend [32,78]. Future ML will be more robust to noise, more capable of learning from smaller or imbalanced datasets, and designed to shift away from binary (yes, no) outcomes to also communicate uncertainty [41,82,102].

## Discussion

The scientific discourse around the use of ML in medical imaging evolves rapidly. While the promises and potential of ML in imaging has been prominently defined, the current scientific discourse highlights multiple remaining challenges. Undoubtedly, medical images differ significantly from any other type of images. They are highly heterogenous, multidimensional, often complex and subject to high interrater variability. For some medical fields they are scarce and difficult to access. Datasets are often incomplete, of low quality, imbalanced, inadequately labeled or annotated, and contain an abundance of images that are of lower value for ML. Current ecosystems are divided in silos, in which large amounts of images are stored, fully separated and with no interconnectivity channels, further increasing heterogeneity and access costs. In the context of ML, all these challenges above present major hurdles, as they limit access to and the use of images for training and validation purposes.

Many of these challenges identified in the literature go well beyond imaging and broadly apply to medical ML in general. However, with imaging being the largest subdomain of medical ML, it can serve as a good indicator for the state of the broader field. Our findings suggest the line between the strengths and challenges of image-based AI is often blurred, with equity as one example. While some scholars argue that ML will increase access and affordability of healthcare, allowing broader geographic reach, others highlight an algorithmic divide, due for instance to bias in training datasets [49,57,103]. Another example is time efficiency. Despite

the emphasis on the time-saving nature of ML, actual clinical benefits for physicians and patients are debatable. Does the time gained allow for less tedious work and more time available for consultation, or will it rather lead to higher workloads, with a higher number of patients seen in less time [3]?

Trustworthiness, perceived as a driver of acceptance and adoption, is explicitly and implicitly present in the discussion around ML, related to model accuracy, reproducibility, generalizability, and explainability [56,67,104–106]. Each of these aspects is ultimately linked to trust, in line with the emphasis placed on trust by policy and regulatory bodies. The European Commission's Joint Research Centre, for instance, calls for an ethics and trust framework that will enable the safe use of AI and ML, while the US Department of Health and Human Services vows to support AI systems that inspire trust and respect privacy and security [107,108]. The High-level Expert Group on Artificial Intelligence established by the European Commission has issued Ethical Guidelines for Trustworthy AI [109]. While not specific to health-related uses of AI, such guidelines emphasize the centrality of trustworthiness. Yet, some scholars challenge the notion of trust in machines, characterizing hype around ML trustworthiness as inappropriate and ineffective. Trust is something given voluntarily, with the trustor vulnerable and reliant on the goodwill of the trustee and cannot be transferred to a relationship between a human and machine. Instead of developing abstract guidelines that foster trust, it may be wiser to focus on developing reliable algorithms, underpinned by policy and regulation [110].

Our findings confirm a strong emphasis on explainability. Explainability is a major discussion topic in the field of medical AI and is not limited to AI for medical imaging. This corresponds with accumulating policy-level pressure. The European Union's General Data Protection Regulation references transparency requirements and data subjects' right to "meaningful information about the logic behind automated decisions using their data" [106,111–113]. Similarly, in its proposed AI Regulation, the European Commission prescribes that high-risk AI systems (which include clinical-grade ML) "shall be designed and developed in such a way to ensure that their operation is sufficiently transparent to enable users to interpret the system's output and use it appropriately" [113]. Yet, there is wider scepticism about the value and feasibility of explainability [114,115]. Ghassemi and colleagues refer to explainability as false hope, as most models are too complex and multi-dimensional to be explained accurately [106]. While explanation efforts (e.g., heat maps in imaging) may provide broad indications of how an algorithm operates, their contribution to understanding individual decisions is superficial, and in some instances misleading [106]. A further challenge is that such explainability methods lack performance indications and add additional risk for error [106]. For example, heat maps indicate how image regions contribute to an algorithm's decision, but not whether these regions are truly clinically relevant [106]. This discussion raises questions over the requirement of explainability of ML in the clinical setting, with important issues of whether and in which circumstances AI should be explained to patients remaining unsettled [114,115].

While explainability remains an important topic, greater attention should be paid to the assessment of ML algorithms, as proposed by other scholars. Most evaluation methods fail to capture what matters most to patients, i.e., the impact of an algorithm on their care and outcomes [116,117]. Comprehensive evaluation frameworks are needed to capture the performance of an algorithm in the clinical setting, and the care-related actions it triggers [116]. In the diagnostic testing field, such evaluations are classified as phase 5 and 6 studies, primarily consisting of randomized controlled trials [118]. Economic evaluations should be an integral part of such frameworks, assessing cost-effectiveness relative to the standard care [40,50]. Cost-saving is an often-mentioned benefit of ML, with little clinical evidence to support this [87].

The predicted evolution of medical ML towards increasingly complex models also warrants consideration. As more data types are integrated into image recognition tasks, the scope of medical ML is likely to expand from simple binary classification in a diagnostic context, to more subtle forms of prognostic assessment and treatment decision support tools. Performance evaluation, validation, and clinical implementation will likely require more granular guidance, improved standards, and best practices. Integrating large-scale omic data may require *ad hoc* adaptations to account for clinical and regulatory uncertainties linked to its use.

## Broader regulatory implications for medical ML

To keep pace with the rapid progress of ML, we identified an urgent need for comprehensive regulatory and governance models that go beyond abstract principles [60,65,70,105,119]. Abstract regulatory and ethical principles can provide a false sense of safety, and bear the risk of being misused to increase trust and acceptability [119]. Concrete guidelines, sector-specific standards, and clear regulatory requirements are needed to promote responsible innovation in the medical AI [119]. As suggested by Reddy and colleagues as well as the WHO Ethics and Governance of Artificial Intelligence for Health Guidance, such frameworks should be comprehensive and target equity, transparency, trustworthiness, accountability, and openness [105,120]. Interdisciplinarity should take center stage, fostering a dialogue between law and ethics experts, computer scientists, healthcare professionals, and patients [3,119].

The dynamic nature of ML also requires a degree of regulatory flexibility, one that acknowledges uncertainty and embraces change and adaptation [121,122]. In 2019, the US Food and Drug Administration (FDA) proposed a "Predetermined Change Protocol", through which manufacturers can report expected algorithm changes, how these changes will occur, and how continued safety will be ensured [61,123]. The FDA subsequently published the AI/ML Software as a Medical Device Action Plan, vowing to develop the proposed regulatory framework and contribute to good ML practice [124]. Its Pre-Certification Pilot program, targeting the regulation and oversight of software-based medical devices, was developed to cover AI/ML-based products and is an important response to the constantly evolving AI/ML landscape.

Such regulatory approaches are an important paradigm shift towards embracing the dynamic nature of ML without discounting safety or timely technology access [61]. Inevitably, such frameworks will depend on a proactive approach to ethical ML development, such as through wider implementation of ethics by design [119], which ensures that ethical challenges are acknowledged and addressed at each stage of ML development [119]

## Limitations

The findings of the present review should be viewed and interpreted with the following limitations in mind. For the purpose of keeping the volume of eligible studies within a manageable range, we decided to focus on a fraction of the existing literature, mainly addressing the big five chronic conditions. This decision was based on the consideration that the big five chronic conditions cover a large and important fraction of currently existing literature in the field. Nonetheless, that focus provides an incomplete picture of published work.

## Conclusion

Our work systematically and comprehensively captured the latest scientific discourse around the use of ML in medical imaging across the continuum of care for the big five chronic diseases. We found a broad range of reported strengths and limitations, ethical and regulatory challenges, as well as pointers for future trends. Medical images, and the ecosystems they are embedded in, present unique challenges. High heterogeneity, complexity, variations in image

quality, scarcity of well-annotated datasets and access hurdles all limit their use for ML training and validation. Among the more broadly applicable ML challenges, we found that the boundaries between strengths and limitations are often blurred, placing a strong positive emphasis on explainability and trustworthiness, with a largely missing discussion about the specific technical and regulatory challenges surrounding these concepts. The uncertainties around the effective translation of image-based ML tools from proof-of-concept phases to clinical integration are equally emphasized. Further efforts are needed to promote this translation, including multistakeholder engagement that takes into consideration the dynamic and complex ecosystem of technical, ethical, and regulatory constraints. Finally, the future of image-based ML is expected to shift towards multi-source models, combining imaging with an array of other data, in a more open access, and explainable manner.

## Supporting information

**S1 File. Search strategy (Web of Science)**
(DOCX)

**S2 File. List of in included studies**
(XLSX)

**S3 File. Data extraction sheet**
(XLSX)

## Author Contributions

**Conceptualization:** Vasileios Nittas, Milo Alan Puhan, Effy Vayena, Alessandro Blasimme.

**Data curation:** Vasileios Nittas, Paola Daniore, Constantin Landers, Felix Gille, Julia Amann, Shannon Hubbs.

**Formal analysis:** Vasileios Nittas.

**Funding acquisition:** Effy Vayena, Alessandro Blasimme.

**Investigation:** Alessandro Blasimme.

**Methodology:** Vasileios Nittas, Milo Alan Puhan, Effy Vayena, Alessandro Blasimme.

**Project administration:** Effy Vayena, Alessandro Blasimme.

**Resources:** Effy Vayena, Alessandro Blasimme.

**Supervision:** Milo Alan Puhan, Effy Vayena, Alessandro Blasimme.

**Validation:** Felix Gille, Julia Amann, Alessandro Blasimme.

**Writing – original draft:** Vasileios Nittas, Shannon Hubbs.

**Writing – review & editing:** Vasileios Nittas, Paola Daniore, Constantin Landers, Felix Gille, Julia Amann, Shannon Hubbs, Milo Alan Puhan, Effy Vayena, Alessandro Blasimme.

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
