## [Decision Letter · Decision Letter 0]

1 Sep 2022

PDIG-D-22-00090

Beyond high hopes: a scoping review of the 2019-2021 scientific discourse on machine learning in medical imaging

PLOS Digital Health

Dear Dr. Nittas,

Thank you for submitting your manuscript to PLOS Digital Health. After careful consideration, we feel that it has merit but does not fully meet PLOS Digital Health's publication criteria as it currently stands. Therefore, we invite you to submit a revised version of the manuscript that addresses the points raised during the review process.

Please submit your revised manuscript within 60 days Oct 31 2022 11:59PM. If you will need more time than this to complete your revisions, please reply to this message or contact the journal office at digitalhealth@plos.org. Please include the following items when submitting your revised manuscript:

We look forward to receiving your revised manuscript.

Kind regards,

Thomas Schmidt

Academic Editor

PLOS Digital Health

Journal Requirements:

1. Please amend your online detailed Financial Disclosure statement. This is published with the article. It must therefore be completed in full sentences and contain the exact wording you wish to be published.

2. If your research concerns only data provided within your submission, please write “All data are in the manuscript and/or supporting information files.” as your Data Availability Statement.

Additional Editor Comments (if provided):

Thank you for submitting your work to PLOS Digital Health. Again, sorry for the delayed review process. 

Three reviewers have now completed their assessment. As stated by our reviewers your work is relevant and timely in the sense that it seeks to summarize very recent trends and challenges within the field.

However, as also noted by two of our reviewers, we would like to propose a revision of the paper. Primarily to clarify some of the issues highlighted: focus on medical imaging instead of the tendency to gravitate towards medical AI in general, a more thorough elaboration of selected chronic diseases, and preferably a discussion on the impact of the exclusion criteria on your findings.

I hope you will find the comments from our reviewers manageable and I invite you to resubmit your article.

Reviewers' comments:

Reviewer's Responses to Questions

**Comments to the Author**

1. Does this manuscript meet PLOS Digital Health’s publication criteria? Is the manuscript technically sound, and do the data support the conclusions? The manuscript must describe methodologically and ethically rigorous research with conclusions that are appropriately drawn based on the data presented.

Reviewer #1: Yes

Reviewer #2: Yes

Reviewer #3: Yes

2. Has the statistical analysis been performed appropriately and rigorously?

Reviewer #1: N/A

Reviewer #2: N/A

Reviewer #3: N/A

3. Have the authors made all data underlying the findings in their manuscript fully available (please refer to the Data Availability Statement at the start of the manuscript PDF file)?

Reviewer #1: Yes

Reviewer #2: Yes

Reviewer #3: Yes

4. Is the manuscript presented in an intelligible fashion and written in standard English?

Reviewer #1: Yes

Reviewer #2: Yes

Reviewer #3: Yes

5. Review Comments to the Author

Reviewer #1: Thank you for this interesting and important topic, the well-written manuscript which greatly adds to the available literature. I have some minor concerns for the authors to consider:

1. In table 1 you included studies “Published after the 01.08.2019 and before 01.09.2021” for the first minute I was confused and thought that the first date was Jan 8th, I would request changing the date form.

2. Give percentages for your results, like:

Line 194 “Almost all publications (n=110)…..”

Line 204 “Most publications (n=102)…..”

Reviewer #2: The manuscript by Nittas et al is a scoping review of the scientific discourse around machine learning in clinical imaging. It identifies the promises and roadblocks that are put forward in published reviews of the field. I do not have much criticism to make. The manuscript is pleasant to read. With regards to the content, I trust the authors to have done a good job summarizing the selected articles. My biggest concern is that of sampling bias: how to ensure that the articles reflect well the points of view of the different scientific communities involved.

Along this line, my most important concern is why limit to review with a focus on one of 5 chronic conditions? 56 reports were excluded because they did not focus on these conditions. I do not understand the logic. Rather, I see this restriction as a bias against broader reviews on the state of AI in health.

Another major concern that I have is that focus on the terms "AI" and "ML". Indeed, biostatisticians use the terms "prediction models" where others would use the term "ML". There is a gradient of techniques between good old statistical models (such as linear models) and more complex AI tools (such as deep learning). Linear models are still in many cases very relevant and can be seen as machine learning tools (they are studied by the machine learning research community). I note that 223 articles were excluded because of lack of focus on ML. Is this adding a bias against the vocabulary used by traditional biostatisticians, and thus downplaying the view of an important subcommunity?

Gaël Varoquaux

Reviewer #3: The authors have provided a timely and needed scoping review of machine learning (ML) techniques in medical imaging. The strengths and promises of ML were well articulated in table 2 and the domain and subdomain separation of utility is well thought out. 

Unfortunaely the biggest drawback for the manuscript in current form is that most of the discussion is general to ML and very few insights about ML applied to medical imaging.

1.I did not get a clear sense of what imaging modalities are under discussion? The state of utilizing ML in modalities that are standard of care such as CT or ultrasound is different from say molecular imaging or magnetic resonance imaging? A section or sub-section dicussing what modalities are under the scope of this review will be very helpful. 

2. The authors searched for five specific chronic conditions of cardiovascular disease , diabetes, stroke, chronic respiratory disease and cancer. But the authors do not comment on what differences did they find in these medical subfields. Are certain fields utilizing these techniques more ? Is it more useful to be thinking of ML in oncology where lesions fall under the “classification” problems than say in diabates where one if more dependent on blood analysis than imaging?

3. I also did not get a good sense of which ML algorithms are being used for in imaging data? Automated classification and regression are the main ML contributions. Is the literature more skewed towards classification tasks? Or is ML being utilized for predictions? 

4. All challenges listed in table 4 are applicable to all ML applications not just imaging applications? What is social bias in the context of medical imaging? 

5.Future directions - Once again, the focus is on general ML vs. imaging specific literature. Some questions that maybe answered in this section are- Do we have a sense of which imaging modalities are better suited to be married to physiological or genomic information? Imaging ia more expensive but provides higher resolution insight than large amounts of physiological data from wearable sensors. What trade offs do the experts see in terms of utilizing ML on imaging vs physiological data? 

5.The authors conclude -"Finally, the future of image-basedML is expected to shift towards multi-source, open access, and explainable tools, with mobile and wearables devices playing an increasingly important role”. Mobile and wearable devices are currently mostly providing physiological data and not imaging data. The papers that the authors refer to (29, 38) seem to refer to personalized physiological data. Please clarify if the review found any instances of personalized imaging being utilized for medical purposes. 

I commend the authors for taking on a much needed scoping review.In its current form the review is more a scope of ML in general and most of those issues are well known to the ML community. It will be very helpful to refocus on specifics of ML in medical imaging.

6. PLOS authors have the option to publish the peer review history of their article (what does this mean?). If published, this will include your full peer review and any attached files.

**Do you want your identity to be public for this peer review?** For information about this choice, including consent withdrawal, please see our Privacy Policy.

Reviewer #1: Yes: Nisreen Al Jallad

Reviewer #2: Yes: Gaël Varoquaux

Reviewer #3: No

---

## [Decision Letter · Decision Letter 1]

2 Jan 2023

Beyond high hopes: a scoping review of the 2019-2021 scientific discourse on machine learning in medical imaging

PDIG-D-22-00090R1

Dear Dr. Nittas,

We are pleased to inform you that your manuscript 'Beyond high hopes: a scoping review of the 2019-2021 scientific discourse on machine learning in medical imaging' has been provisionally accepted for publication in PLOS Digital Health.

Best regards,

Thomas Schmidt

Academic Editor

PLOS Digital Health

Dear authors

Thank you for resubmitting your manuscript to PLOS Digital Health. Apologies for the delay in processing. Thank you for carefully addressing the comments of our reviewers. Your revisions have improved the final manuscript, and I have no quarrels recommending your work for publication in our journal.

Reviewer Comments (if any, and for reference):

Reviewer's Responses to Questions

**Comments to the Author**

1. If the authors have adequately addressed your comments raised in a previous round of review and you feel that this manuscript is now acceptable for publication, you may indicate that here to bypass the “Comments to the Author” section, enter your conflict of interest statement in the “Confidential to Editor” section, and submit your "Accept" recommendation.

Reviewer #3: All comments have been addressed

2. Does this manuscript meet PLOS Digital Health’s publication criteria? Is the manuscript technically sound, and do the data support the conclusions? The manuscript must describe methodologically and ethically rigorous research with conclusions that are appropriately drawn based on the data presented.

Reviewer #3: Yes

3. Has the statistical analysis been performed appropriately and rigorously?

Reviewer #3: N/A

4. Have the authors made all data underlying the findings in their manuscript fully available (please refer to the Data Availability Statement at the start of the manuscript PDF file)?

Reviewer #3: Yes

5. Is the manuscript presented in an intelligible fashion and written in standard English?

Reviewer #3: Yes

6. Review Comments to the Author

Reviewer #3: I applaud the authors for taking the time and effort to revise the manuscript and realign it to medical imaging applications. I especially appreciate the clarity that added table (table 3) discussing ML challenges in medical imaging provides to readers.

7. PLOS authors have the option to publish the peer review history of their article (what does this mean?). If published, this will include your full peer review and any attached files.

**Do you want your identity to be public for this peer review?** For information about this choice, including consent withdrawal, please see our Privacy Policy.

Reviewer #3: No
